# OpenReview forum: "Safe-LLaVA: A Privacy-Preserving Vision-Language Dataset and Benchmark for Biometric Safety"
_ICLR.cc/2026/Conference — ICLR 2026 Conference Withdrawn Submission_

### Official Review · Reviewer_RdD4 · 2025-10-27

**Soundness:** 2
**Presentation:** 2
**Contribution:** 2
**Rating:** 2
**Confidence:** 4

**Summary:**

This paper studies how well current VLMs protect against the leakage of sensitive biometric information contained in input images and methods to train VLMs such that they are more secure and private in this regard. The authors' contributions are two-fold. First, they curate PRISM, a VQA dataset meant to evaluate how often VLMs leak sensitive biometric information. PRISM consists of nearly 30K questions across 5 attribute types. The authors then detail the creation of Safe-LLaVA: a filtered version of the original LLaVA pretraining and instruction tuning datasets with biometric information removed or abstracted using an LLM-in-the-loop pipeline using GPT-4o as a judge. The authors pretrain and instruction fine-tune a base LLaVA architecture on this dataset and, through evaluation on the constructed PRISM dataset, find that it achieves the best biometric data protection compared to other off-the-shelf models without severely degrading model utility (performance on common VQA benchmarks).

**Strengths:**

1. The paper provides a valuable contribution in creating Safe-LLaVA, which is a safe, filtered version of the original LLaVA training dataset to protect biometric data. While the authors do acknowledge that they are not the first work to filter pretraining and fine-tuning corpora based on safety criteria, Safe-LLaVA could serve as an important and useful artifact for use by the community since it uses a more sophisticated LLM-as-a-judge pipeline for filtering.
2. The authors also demonstrate that pretraining and fine-tuning on their filtered data do improve performance on their created PRISM dataset when compared to off-the-shelf models.
3. The paper also has a useful privacy-utility analysis, where they find that their model is able to maintain privacy without sacrificing utility. This is an important characteristic for VLMs, as otherwise the model would be useless to humans. Additionally, the authors also find that the model trained on Safe-LLaVA is truly robust to adversarial and more explicit queries via "hard prompts".

**Weaknesses:**

1. I generally do not agree with the motivation for this work. Why would it not be okay for VLMs to disclose some biometric data? For instance, in Figure 1, a human can clearly identify the two mentioned biometric data points from the image. What harm is there in a VLM confirming what a human can already clearly see? Specifically, as you note in the introduction, GDPR is against the *unauthorized use of SCPD*, but it does not seem that *simply identifying* these traits is harmful. I think that if it can be shown that the proposed training procedure prevents VLMs from *making decisions based on these attributes*, this would be an important contribution.

2. Some basic baselines are missing. For instance, models can be simply prompted and told not to reveal each specific type of biometric data - potentially few-shot examples could also be provided. This is a simple baseline that is also much more computationally feasible. I suspect prompting might be highly effective since Chen et al., 2023 show that this is a useful approach. Additionally, it would be helpful to see the evaluation of large closed-source models like GPT-40, GPT-5, Gemini, Claude, etc., on PRISM. An evaluation on at least a subset of these models would be needed to calibrate the results of the improvement when training on Safe-LLaVA.

3. Analysis of adapting to new classes of data that may be classified as protected in the future is missing. For instance, the authors could run an experiment where they train on only 4 of the 5 biometric categories and test on the held-out fifth.

**Questions:**

1. What was the relative effect of instruction-tuning vs. pretraining? Specifically, could we get a similar effect by doing the instruction tuning step only, instead of pretraining the model again? It would seem like we would only need a limited dataset of a few thousand examples to tune the model on to learn the abstention/abstraction behavior.

2. Could we use a weaker model / string matching rules to do the filtering for Safe-LLaVA? It seems like this is not too complicated or reasoning-intensive a task.

3. How does the model respond to queries just outside the privacy barrier? For instance, consider a question like "What is the color of the eyes of the teddy bear?". Clearly, this is a benign question; however, a model trained with Safe-LLaVA may potentially refuse to answer.

**Details Of Ethics Concerns:**

I am concerned about the use of ```icrawler``` to collect images for the PRISM dataset. Do you have the proper license for these images? Can this dataset be redistributed upon publication?

---

### Official Review · Reviewer_i23i · 2025-10-29

**Soundness:** 3
**Presentation:** 3
**Contribution:** 3
**Rating:** 4
**Confidence:** 3

**Summary:**

The paper argues that multimodal LLMs (MLLMs) frequently infer or reveal sensitive biometric attributes even when not asked, and that there is no standard way to measure or mitigate this risk today. It proposes 3 main contributions: (1) PRISM, a benchmark to evaluate both explicit refusal of biometric questions and implicit leakage in open-ended descriptions; (2) Safe-LLaVA, a privacy-preserving training corpus produced by systematically removing explicit and implicit biometric content from LLaVA; (3) Models trained with Safe-LLaVA achieve significant improvements in privacy-related safety.

**Strengths:**

- Clear scope. The paper focuses squarely on biometric privacy and separately probes explicit refusal and implicit leakage with both direct and open-ended prompts, which is closer to real usage than refusal-only tests.
- Strong evaluation results. The experiments show that PRISM can reveal existing model biometric safety risks, and that the Safe-LLaVA dataset and the corresponding fine-tuned models can significantly improve privacy safety.

**Weaknesses:**

- Too many missing details. The main text omits many necessary details, for example, the rationale for the five PRISM categories, the PRISM data schema, whether there is class imbalance, and specifics of SAFE-LLaVA model training. I understand the space constraints, but the corresponding sections should at least mention these concepts and provide links to the appendix.
- Limited model variety. Although the experiments are detailed, more than 60% of the evaluated models belong to the LLaVA family, and the privacy-hardening experiments are conducted only on the LLaVA architecture.

**Questions:**

1. GDPR motivation. The authors repeatedly cite the GDPR as motivation, but they do not clearly map the paper’s stated risks to specific GDPR provisions. For example, under which section does the GDPR explicitly prohibit outputting any information about eye color, age, and body weight? When a user uploads an image and asks related questions, which GDPR clauses would this implicate?

2. PRISM Data balance & bias. What are the per-class counts and demographic distributions? Any steps to mitigate imbalance or over-representation? In my view, eye color, age, and weight are fine-grained attributes and may not be appropriate to place at the same level as race.

3. Performance of other MLLMs on PRISM, especially commercial models. How do non-LLaVA and closed-source model perform under the PRISM benchmark?

4. Architecture generalization. Can Safe-LLaVA fine-tuning replicate gains on non-LLaVA backbones (e.g., InternVL, Qwen-VL, MiniGPT-4) to demonstrate broader applicability?

5. Minor — refusal modes. When the model refuses, are there distinct refusal types? For example, for weight-related questions, does it sometimes respond with an “unable to determine” style refusal rather than a privacy-motivated refusal?

**Details Of Ethics Concerns:**

Recommendation: add an ETHICAL STATEMENT section to discuss compliance considerations during dataset construction.

---

### Official Review · Reviewer_TmZf · 2025-10-31

**Soundness:** 3
**Presentation:** 3
**Contribution:** 2
**Rating:** 4
**Confidence:** 4

**Summary:**

This paper proposes PRISM, a benchmark designed to evaluate MLLMs on their ability to (1) refuse biometric-related prompts and (2) suppress biometric leakage in open-ended responses while maintaining semantic fidelity. Moreover, Safe-LLaVA is conducted by removing explicit and implicit biometric cues from captions, questions and answers, forming the first privacy-preserving MLLM training data.

**Strengths:**

The paper proposes a benchmark to assess the risk of biometric information leakage by models. This benchmark introduces a novel risk quantification metric for implicit evaluation, which measures the potential leakage risk of biometric information at both the attribute level and the sentence level. Additionally, the proposed Safe-LLaVA training set can partially mitigate a model’s tendency to leak biometric information.

**Weaknesses:**

•	The paper selects five biometric information types as the privacy evaluation dimensions, but does not provide justification for choosing these specific types.. Intuitively, attributes such as eye color may not be strongly associated with severe privacy violations, raising the question of the practical importance of preventing models from outputting such information. Furthermore, the paper utilizes foundation models such as Qwen2.5-VL and GPT-4o to recognize biometric information in images for data distribution statistics and masking, which implies that strong biometric attribute recognition capabilities can be valuable. The authors should discuss under what circumstances a model should avoid outputting biometric information, and construct concrete risk scenarios accordingly.

•	The proposed PRISM benchmark lacks a clear comparison with relevant existing work, such as quantitative metrics including dataset scale and evaluation dimensions.

•	Safe-LLaVA mitigates sensitive information leakage by removing or replacing biometric attributes in the dataset. However, this approach may lead to over-protection. For instance, the model might also refuse benign requests such as asking about the eye color or weight of animals, but the paper does not address this potential drawback.

**Questions:**

•	Constructing Safe-LLaVA appears to require extensive access to GPT-4o. What is the estimated time and computational/financial cost?

•	The paper relies heavily on large models for evaluation, but does not report the accuracy of these model-based evaluations. A comparison between model judgments and human assessment accuracy should be provided.

•	The authors state that PRISM underwent manual filtering to ensure image quality. What were the filtering criteria, and how many images were removed?

---

### Official Review · Reviewer_Cbku · 2025-10-31

**Soundness:** 3
**Presentation:** 3
**Contribution:** 3
**Rating:** 6
**Confidence:** 4

**Summary:**

This paper addresses privacy concerns (i.e. biometric attributes) in Multimodal Large Language Models (MLLMs). It introduces: 1) PRISM - a benchmark for evaluating biometric leakage covering 5 attributes (race, gender, age, eye color, body weight) across 22 sub-categories with 2,200 images and 28.6k question-answer pairs, 2) Safe-LLaVA - the first privacy-preserving MLLM training dataset created by systematically removing biometric information from the original LLaVA dataset using GPT-4o. And furthermore, this work demonstrates that models trained on Safe-LLaVA achieve substantially reduced biometric leakage (>90% protection scores) while maintaining performance on general benchmarks.

**Strengths:**

1. The paper tackles an important privacy issue in MLLMs, it provides a benchmark for evaluation (PRISM) and a dataset (Safe-LLaVA), addressing the problem from multiple angles.

2. This paper conducts extensive experiments with two evaluators (GPT and Gemini), diverse metrics (explicit refusal accuracy, implicit leakage at attribute and sentence levels), and comparisons across multiple public state-of-the-art MLLMs. It demonstrates that Safe-LLaVA models achieve 98%+ protection scores.

3. Potentially, these are good dataset, benchmark for the community.

**Weaknesses:**

1. Generalization concern: all experiments in the paper are based on LLaVA models. It's unclear how well the approach generalizes to other MLLM architectures or training paradigms.

2. Heavily rely on GPT-4o and Gemini: both the evaluation and dataset curation process heavily rely on these closed frontier models. May have potential issue for scaling up, both evaluation scale and dataset scale.

**Questions:**

1. Cross-dataset generalization: have you tested trained Safe-LLaVA models on other privacy benchmarks? How does performance transfer?

2. Table 4 shows some degradation in refusal accuracy under hard prompts (e.g., eye color drops from ~95% to ~81-92% for Safe-LLaVA 0.5B -> 7B). What accounts for this gap?

---

### Note · Authors · 2025-11-14

I have read and agree with the venue's withdrawal policy on behalf of myself and my co-authors.